# BEiT v2: Masked Image Modeling with Vector-Quantized Visual Tokenizers

## Abstract

Masked image modeling (MIM) has demonstrated impressive results in self-supervised representation learning by recovering corrupted image patches. However, most existing studies operate on low-level image pixels, which hinders the exploitation of high-level semantics for representation models. In this work, we propose to use a semantic-rich visual tokenizer as the reconstruction target for masked prediction, providing a systematic way to promote MIM from pixel-level to semantic-level. Specifically, we propose vector-quantized knowledge distillation to train the tokenizer, which discretizes a continuous semantic space to compact codes. We then pretrain vision Transformers by predicting the original visual tokens for the masked image patches. Furthermore, we introduce a patch aggregation strategy which associates discrete image patches to enhance global semantic representation. Experiments on image classification and semantic segmentation show that BEiT v2 outperforms all compared MIM methods. On ImageNet-1K (224 size), the base-size BEiT v2 achieves $85.5\%$ top-1 accuracy for fine-tuning and $80.1\%$ top-1 accuracy for linear probing. The large-size BEiT v2 obtains $87.3\%$ top-1 accuracy for ImageNet-1K (224 size) fine-tuning, and $56.7\%$ mIoU on ADE20K for semantic segmentation. The code can be found in the supplementary materials.

## 1 Introduction

Masked image modeling (MIM), which greatly relieves the annotation-hungry issue of vision Transformers, has demonstrated great potential in learning visual representations (Bao et al., 2022; He et al., 2022). Given an image, the pretraining objective of MIM is to recover the masked patches so that rich context information is captured by the representation model. Taking BEiT (Bao et al., 2022) as an example, each image has two views during pretraining, *i.e.*, image patches, and visual tokens. The original image is first tokenized to discrete tokens. Randomly sampled image patches are then masked before being fed to vision Transformers. The pretraining objective is to recover the original visual tokens based on the corrupted image patches. The pretrained vision encoder can be deployed and finetuned on various downstream tasks by appending lightweight task layers.

Existing MIM approaches can be coarsely categorized to three according to the reconstruction targets: low-level image elements (*e.g.*, raw pixels; He et al. 2022; Fang et al. 2022; Liu et al. 2022), hand-crafted features (*e.g.*, HOG features; Wei et al. 2021), and visual tokens; Bao et al. 2022; Wang et al. 2022; Dong et al. 2021; El-Nouby et al. 2021; Chen et al. 2022. However, all the reconstruction targets are about, explicitly or implicitly, low-level image elements while underestimating high-level semantics. In comparison, the masked words in language modeling (Devlin et al., 2019) are all about high-level semantics, which motivates us to tap the potential of MIM by exploiting semantic-aware supervision during pretraining.

In this work, we propose a self-supervised representation learning approach, termed BEiT v2, with the aim to improve MIM pretraining by constructing a semantic-aware visual tokenizer. Our approach is developed on the BEiT method which is simple yet effective. The novelty lies in introducing the Vector-Quantized Knowledge Distillation (VQ-KD) algorithm to discretize a semantic space. The VQ-KD encoder first converts the input image to discrete tokens according to a learnable codebook. The decoder then learns to reconstruct the semantic features encoded by a teacher model, conditioning on the discrete tokens. After training VQ-KD, its encoder is used as a semantic visual tokenizer for BEiT pretraining, where the discrete codes serve as supervision signals.

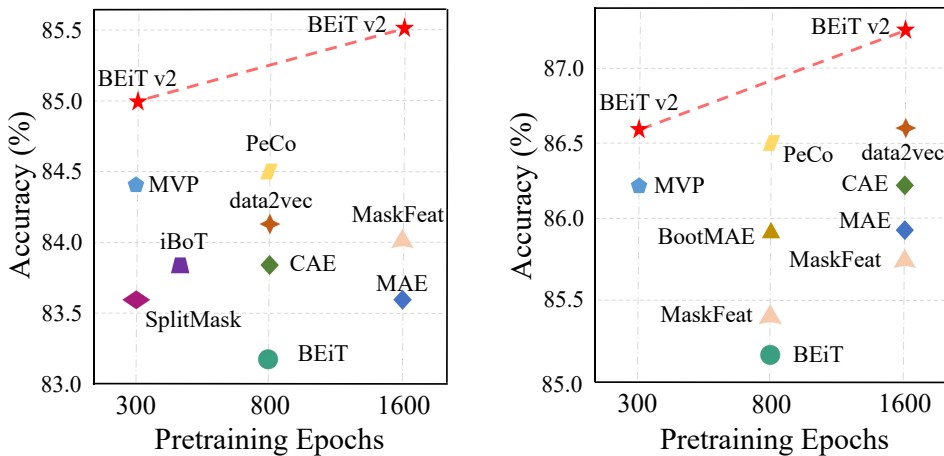

Figure 1: Top-1 fine-tuning accuracy on ImageNet (224 size). **Left**: ViT-B/16. **right**: ViT-L/16.

Considering the discreteness of tokens, we further introduce a patch aggregation strategy which explicitly encourages the `[CLS]` token to associate all patches (Gao & Callan, 2021). Such a strategy resolves the issue that MIM put patch reconstruction the first place which diminishes learning global image representations. As a result, BEIT V2 improves the capacity of learned image representation, as supported by the linear probing experiments. Moreover, the enhanced representations also boosts the performance of other tasks.

We conduct self-supervised learning on ImageNet-1k for both base- and large-size vision Transformers, which are evaluated on downstream tasks, *e.g.*, image classification, linear probing, and semantic segmentation. As shown in Figure 1, BEIT V2 outperforms previous self-supervised learning algorithms by a large margin on ImageNet fine-tuning, *e.g.*, improving over BEIT (Bao et al., 2022) by about two points for both ViT-B/16 and ViT-L/16. BEIT V2 outperforms all compared MIM methods on ImageNet linear probing while achieving large performance gains on ADE20k for semantic segmentation.

The contributions of this work are summarized as follows:

- We propose vector-quantized knowledge distillation, promoting masked image modeling from pixel-level to semantic-level for self-supervised representation learning.

- We introduce a patch aggregation strategy, which enforces global structure given discrete semantic tokens, and improves the performance of learned representations.

- We conduct extensive experiments on downstream tasks including ImageNet fine-tuning, linear probing, and semantic segmentation. Experimental results show that the proposed approach significantly improves performance across model sizes, training steps, and downstream tasks.

## 2 METHODOLOGY

BEIT V2 inherits the masked image modeling framework defined by BEIT (Bao et al., 2022), which uses a visual tokenizer to convert each image to a set of discrete visual tokens. The training target is to recover the masked visual tokens, each of which corresponds to an image patch. In Section 2.2, we introduce a vector-quantized knowledge distillation algorithm, which is used to train a visual tokenizer. In Section 2.3, we employ the visual tokenizer for BEIT pretraining with the help of the patch aggregation strategy.

### 2.1 IMAGE REPRESENTATION

The vision Transformers (ViTs; Dosovitskiy et al. 2020) are employed as the backbone networks to obtain image representations. The input image $x \in \mathbb{R}^{H \times W \times C}$ is reshaped to $N = HW/P^2$ patches

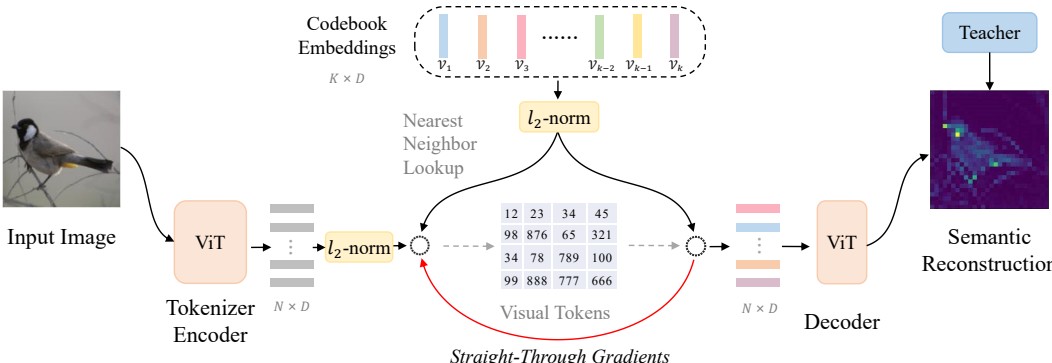

Figure 2: Pipeline for visual tokenizer training. After training, each image is converted to discrete visual tokens.

$\{\boldsymbol{x}_i^p\}_{i=1}^N$, where $\boldsymbol{x}^p \in \mathbb{R}^{N \times (P^2 C)}$ and $(P, P)$ is the patch size. In experiments, each $224 \times 224$ image is split to a $14 \times 14$ grid of image patches, where each patch is $16 \times 16$. The image patches $\{\boldsymbol{x}_i^p\}_{i=1}^N$ are then flattened and linearly projected to input embeddings for Transformers. The encoding vectors are denoted as $\{\boldsymbol{h}_i\}_{i=1}^N$, which corresponds to $N$ image patches.

## 2.2 TRAINING VISUAL TOKENIZER

We propose vector-quantized knowledge distillation (VQ-KD) to train the visual tokenizer, Figure 2, where the visual tokenizer and the decoder are two vital modules.

The visual tokenizer maps an image to a sequence of visual tokens, *a.k.a.*, discrete codes. To be specific, an image $\boldsymbol{x}$ is tokenized to $\boldsymbol{z} = [z_1, z_2, \cdots, z_N] \in \mathcal{V}^{(H/P) \times (W/P)}$, where the visual vocabulary (*a.k.a.*, codebook) $\mathcal{V} \in \mathbb{R}^{K \times D}$ contains $K$ discrete codebook embeddings.

The tokenizer is consist of a vision Transformer encoder, and a quantizer. The tokenizer first encodes the input image to vectors. Then, the vector quantizer looks up the nearest neighbor in the codebook for each patch representation $\boldsymbol{h}_i$. Let $\{\boldsymbol{v}_1, \boldsymbol{v}_2, \cdots, \boldsymbol{v}_K\}$ denote the codebook embeddings. For the $i$-th image patch, its quantized code is calculated as

$$\boldsymbol{z}_i = \arg\min_j ||\ell_2(\boldsymbol{h}_i) - \ell_2(\boldsymbol{v}_j)||_2, \tag{1}$$

where $j \in \{1, 2, \cdots, K\}$ and $\ell_2$ normalization is used for codebook lookup (Yu et al., 2021). The above distance is equivalent to finding codes according to cosine similarity.

After quantizing the image to visual tokens, we feed the $\ell_2$-normalized codebook embeddings $\{\ell_2(\boldsymbol{v}_{z_i})\}_{i=1}^N$ to the decoder. The decoder is also a multi-layer Transformer. The output vectors $\{\boldsymbol{o}_i\}_{i=1}^N$ aim at reconstructing the semantic features of a teacher model, *e.g.*, DINO (Caron et al., 2021), and CLIP (Radford et al., 2021). Let $\boldsymbol{t}_i$ denote the teacher model's feature vector of the $i$-th image patch. During training, we maximize the cosine similarity between the decoder output $\boldsymbol{o}_i$ and the teacher guidance $\boldsymbol{t}_i$.

Because the quantization process (Equation 1) is non-differentiable, the gradients are directly copied from the decoder input to the encoder output (van den Oord et al., 2017), Figure 2, to back-propagate gradients to the encoder. Intuitively, the quantizer looks up the nearest code for each encoder output, while the gradients of codebook embeddings indicate useful optimization directions for the encoder.

The training objective of VQ-KD is defined as

$$\max \sum_{x \in \mathcal{D}} \sum_{i=1}^N \cos(\boldsymbol{o}_i, \boldsymbol{t}_i) - ||\text{sg}[\ell_2(\boldsymbol{h}_i)] - \ell_2(\boldsymbol{v}_{z_i})||_2^2 - ||\ell_2(\boldsymbol{h}_i) - \text{sg}[\ell_2(\boldsymbol{v}_{z_i})]||_2^2, \tag{2}$$

where $\text{sg}[\cdot]$ stands for the stop-gradient operator which is an identity at the forward pass while having zero gradients during the backward pass. $\mathcal{D}$ represents the image data used for tokenizer training.

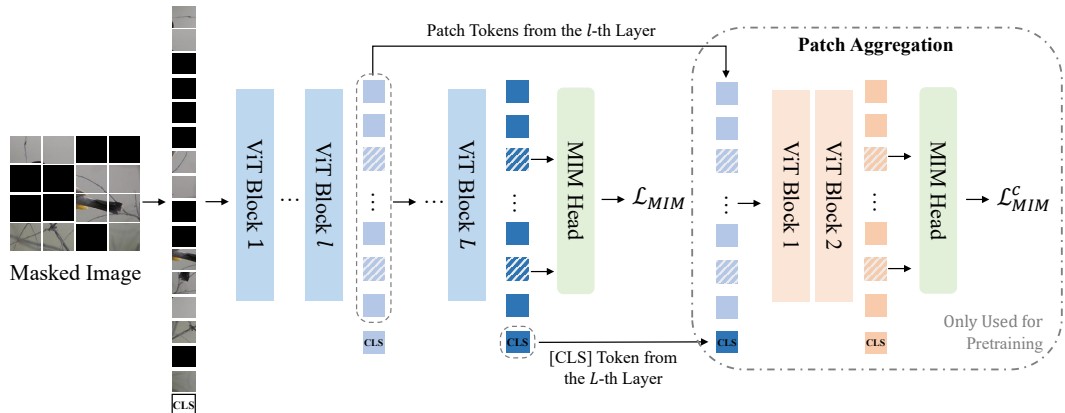

Figure 3: The MIM framework equipped with patch aggregation. The pretraining loss is the summation of $\mathcal{L}_{\text{MIM}}$ and $\mathcal{L}_{\text{MIM}}^c$. The loss term $\mathcal{L}_{\text{MIM}}^c$ explicitly encourages the `[CLS]` token to aggregate patch information to global representations.

**Improving codebook utilization.** A common issue of vector quantization training is codebook collapse. In other words, only a small proportion of codes are used. Empirical strategies (van den Oord et al., 2017; Yu et al., 2021) can be used to alleviate this issue. Equation 1 shows that we compute the $\ell_2$-normalized distance to find the nearest code while reducing the dimension of codebook embedding space to 32-d. The low-dimensional codebook embeddings are mapped back to higher-dimensional space before being fed to the decoder. Exponential moving average (van den Oord et al., 2017) is employed to update the codebook embeddings. Exponential moving average tends to be more stable for VQ-KD training.

## 2.3 Pretraining BEiT v2

We follow the MIM setup in BEiT (Bao et al., 2022) to pretrain vision Transformers for image representations. Given an input image $x$, around 40% image patches are block-wisely chosen and masked. The masked position is termed as $\mathcal{M}$. Then, a shared learnable embedding $e_{[\text{M}]}$ is used to replace the original image patch embeddings $e_i^p$ if $i \in \mathcal{M}$: $x_i^{\mathcal{M}} = \delta(i \in \mathcal{M}) \odot e_{[\text{M}]} + (1 - \delta(i \in \mathcal{M})) \odot x_i^p$, where $\delta(\cdot)$ is the indicator function. Subsequently, we prepend a learnable `[CLS]` token to the input, *i.e.*, $[e_{\text{CLS}}, \{x_i^{\mathcal{M}}\}_{i=1}^N]$, and feed them to the vision Transformer. The final encoding vectors are denoted as $\{h_i\}_{i=0}^N$, where $h_0$ is for the `[CLS]` token.

Next, we instantiate the MIM head as a simple fully-connection layer, and then use it to predict the visual tokens of the masked positions based on the corrupted image $x^{\mathcal{M}}$. For each masked position $\{h_i : i \in \mathcal{M}\}_{i=1}^N$, a softmax classifier predicts the visual tokens $p(z_i | h_i) = \text{softmax}_{z_i}(W_c h_i + b_c)$, where $W_c, b_c$ respectively mean weights and biases of the MIM head. The visual tokens are obtained by the tokenizer trained in Section 2.2, which provides supervisions for the MIM self-supervised learning procedure. The training loss of MIM is defined as

$$\mathcal{L}_{\text{MIM}} = -\sum_{x \in \mathcal{D}} \sum_{i \in \mathcal{M}} \log p(z_i | x_i^{\mathcal{M}}), \qquad (3)$$

where $z_i$ denotes the visual tokens of the original image, and $\mathcal{D}$ the pretraining images. Notice that the number of visual tokens is the same as the number of image patches in this work.

**Pretraining global representation.** Inspired by (Gao & Callan, 2021), we pretrain the `[CLS]` token for global image representation. The goal is to mitigate the discrepancy between patch-level pretraining and image-level representation aggregation. As illustrated in Figure 3, a representation bottleneck is constructed to encourage the `[CLS]` token to gather information as much as possible. For a $L$-layer Transformer, let $\{h_i^l\}_{i=1}^N$ denote the $l$-th layer's output vectors, where $l \in \{1, 2, \cdots, L\}$. To pretrain the last layer's `[CLS]` token $h_{\text{CLS}}^L$, we concatenate it with the intermediate $l$-th layer's patch vectors $\{h_i^l\}_{i=1}^N$, *i.e.*, $S = [h_{\text{CLS}}^L, h_1^l, \cdots, h_N^l]$. We then feed $S$ to a shallow (*e.g.*, two layers)

Transformer decoder and conduct masked prediction again, *i.e.*, $p(\boldsymbol{z}|\boldsymbol{S}) = \mathrm{softmax}_{\boldsymbol{z}}(\boldsymbol{W}_c\boldsymbol{S} + \boldsymbol{b}_c)$. Notice that the parameters are shared for both MIM heads and the MIM loss is also computed at mask positions as in Equation 3. Accordingly, the final training loss is defined as the summation of two terms, *i.e.*, the original loss at the $L$-th layer, and the shallow Transformer decoder's MIM loss. Overall framework refers to Appendix C.

Intuitively, the model favors pushing the global information to $\boldsymbol{h}_{\mathtt{CLS}}^L$, because the model tends to fully utilize the parameters from $(l + 1)$-th layer to $L$-th layer, to decrease the additional MIM loss. The information-flow bottleneck encourages the [CLS] token towards more reliable global representations than its untrained counterparts. Moreover, the enhanced representations also facilitate various downstream tasks. Notice that the newly added shallow decoder is only used to pretrain the [CLS] token, which is discarded after pretraining.

## 3 EXPERIMENTS

The pretrained models are evaluated on image classification and semantic segmentation tasks. For image classification, the models are trained on ImageNet-1K (Russakovsky et al., 2015) and evaluated by (1) top-1 accuracy about fine-tuning and (2) top-1 accuracy about linear probing (only fine-tuning the classification head). For semantic segmentation, experiments are conducted on the ADE20K dataset (Zhou et al., 2019) and the performance is evaluated using the mIoU protocol.

### 3.1 PRETRAINING SETUP

**Visual tokenizer training.** We instantiate the visual tokenizer of VQ-KD as ViT-B/16 for both base- and large-size BEiT v2 pretraining. The decoder network is a three-layer standard Transformer, which has the same dimension and number of attention heads as the tokenizer encoder. The OpenAI CLIP-B/16 (Radford et al., 2021) is employed as the teacher model and train VQ-KD on ImageNet-1k with 224×224 resolution. Notice that we use the same base-size teacher to train the visual tokenizer for both base- and large-size pretraining. The code size $K$ is set as 8192 and code dimension $D$ as 32 by default. Refer to Appendix D for more training details.

**Masked image modeling.** We follow the settings used in BEiT (Bao et al., 2022) pretraining and use ImageNet-1K without labels as the pretraining data for self-supervised learning. The input image resolution is set as 224x224 during pretraining. The pretrained base- and large-size vision Transformers (Dosovitskiy et al., 2020) with $16 \times 16$ patch size are denoted as ViT-B/16 and ViT-L/16, respectively. For the patch aggregation strategy, we set $l = 9$ for ViT-B/16, $l = 21$ for ViT-L/16, and the depth as 2 by default. A block-wise masking mechanism is adopted under the mask ratio of 40% (*i.e.*, about 75 image patches). More pretraining details can be found in Appendix E.

### 3.2 IMAGE CLASSIFICATION

Both the fine-tuning accuracy and linear probing accuracy are evaluated on ImageNet-1k by default. The models are also evaluated on several ImageNet variants to demonstrate their favorable generalization ability.

**Fine-tuning setup.** We follow the protocol proposed in BEiT (Bao et al., 2022) to fine-tune the pretrained BEiT v2 model (see Appendix F for more details). In Table 1, we report the top-1 fine-tuning accuracy results and compare BEiT v2 with recent MIM methods.

From Table 1, base-size BEiT v2 with a 300-epoch pretraining schedule reaches 85.0% top-1 accuracy, which outperforms BEiT, CAE, SplitMask and PeCo by 2.1%, 1.4%, 1.4% and 0.9% respectively. Compared with masked distillation methods, like MVP, BEiT v2 also shows superiority. Furthermore, with a longer pretraining schedule, BEiT v2 achieves 85.5% top-1 accuracy, developing a new state of the art on ImageNet-1K among self-supervised methods. Meanwhile, BEiT v2 using ViT-L/16 with 300 epochs reaches 86.6% top-1 accuracy, which is comparable to data2vec with 1600 epochs. A longer pretraining schedule further boosts the performance to 87.3%.

Following BEiT, we add an intermediate fine-tuning phase between the pretraining stage and the fine-tuning stage. Only the intermediate fine-tuning phase uses the ImageNet-21k dataset. As

Table 1: Fine-tuning results of image classification and semantic segmentation on ImageNet-1K and ADE20k. UperNet (Xiao et al., 2018) is used as the task layer for semantic segmentation with single-scale (512 size) input.

| Methods | Pretraining Epochs | ImageNet Top-1 Accuracy(%) | ADE20k mIoU(%) |
|---|---|---|---|
| *Base-size models (ViT-B/16)* | | | |
| BEɪT (Bao et al., 2022) | 300 | 82.9 | 44.7 |
| CAE (Chen et al., 2022) | 300 | 83.6 | 48.3 |
| SplitMask (El-Nouby et al., 2021) | 300 | 83.6 | 45.7 |
| MaskFeat (Wei et al., 2021) | 300 | 83.6 | N/A |
| PeCo (Dong et al., 2021) | 300 | 84.1 | 46.7 |
| MVP (Wei et al., 2022) | 300 | 84.4 | 52.4 |
| iBoT (Zhou et al., 2022) | 400 | 83.8 | 50.0 |
| **BEɪT v2 (ours)** | 300 | **85.0** | **52.7** |
| *Base-size models (ViT-B/16) + pretrain longer* | | | |
| BEɪT (Bao et al., 2022) | 800 | 83.2 | 45.6 |
| PeCo (Dong et al., 2021) | 800 | 84.5 | 48.5 |
| data2vec (Baevski et al., 2022) | 800 | 84.2 | N/A |
| MAE (He et al., 2022) | 1600 | 83.6 | 48.1 |
| CAE (Chen et al., 2022) | 1600 | 83.9 | 50.2 |
| **BEɪT v2 (ours)** | 1600 | **85.5** | **53.1** |
| + Intermediate fine-tuning with ImageNet-21k | | **86.5** | **53.5** |
| *Large-size models (ViT-L/16)* | | | |
| iBoT (Zhou et al., 2022) | 250 | 84.8 | N/A |
| MaskFeat (Wei et al., 2021) | 300 | 84.4 | N/A |
| MVP (Wei et al., 2022) | 300 | 86.3 | 54.3 |
| **BEɪT v2 (ours)** | 300 | **86.6** | **55.0** |
| *Large-size models (ViT-L/16) + pretrain longer* | | | |
| BEɪT (Bao et al., 2022) | 800 | 85.2 | 53.3 |
| MaskFeat (Wei et al., 2021) | 1600 | 85.7 | N/A |
| MAE (He et al., 2022) | 1600 | 85.9 | 53.6 |
| CAE (Chen et al., 2022) | 1600 | 86.3 | 54.7 |
| data2vec (Baevski et al., 2022) | 1600 | 86.6 | N/A |
| **BEɪT v2 (ours)** | 1600 | **87.3** | **56.7** |
| + Intermediate fine-tuning with ImageNet-21k | | **88.4** | **57.5** |

Table 2: Top-1 accuracy of linear probing on ImageNet-1k. All methods are based on ViT-B/16 pretrained for 300 epochs except MAE for 1600 epochs.

| Methods | Linear Probe |
|---|---|
| BEɪT (Bao et al., 2022) | 56.7 |
| CAE (Chen et al., 2022) | 64.1 |
| MAE (He et al., 2022) | 67.8 |
| MVP (Wei et al., 2022) | 75.4 |
| MoCo v3 (Chen et al., 2021) | 76.7 |
| BEɪT v2 (ours) | **80.1** |

Table 3: Robustness evaluation on three ImageNet variants (Hendrycks et al., 2021b;a; Wang et al., 2019).

| Methods | ImageNet Adversarial | ImageNet Rendition | ImageNet Sketch |
|---|---|---|---|
| *ViT-B/16* | | | |
| MAE | 35.9 | 48.3 | 34.5 |
| BEɪT v2 | **54.4** | **61.0** | **45.6** |
| *ViT-L/16* | | | |
| MAE | 57.1 | 59.9 | 45.3 |
| BEɪT v2 | **69.0** | **69.9** | **53.5** |

shown in Table 1, we find that intermediate fine-tuning achieves about 1% performance gain on image classification for both base- and large-size models. Refer to Appendix B for more results of intermediate fine-tuning.

**Linear probing.** Keeping the backbone model frozen and training a linear classification head atop the image-level representations, linear probing has been a widely considered measure for self-supervised learning. We average the patch tokens as the global representation for the models without

Table 4: Ablation studies under VQ-KD settings. "Base&1x768x12" denotes that the encoder network is ViT-Base while the decoder is a Transformer with depth 1, dimensions 768, and head 12. "Reconst. Loss" is the reconstruction loss of VQ-KD. Reconstruction loss and codebook usage are measured on the validation set. After 300 epochs of pretraining, our method reports the top-1 fine-tuning accuracy and linear probing accuracy on ImageNet-1k, and mIoU on ADE20k. The default setting is highlighted in  gray .

| VQ-KD Architecture | Codebook | Reconst. Loss | Codebook Usage | ImageNet Fine-tuning | ImageNet Linear Probe | ADE20k |
|---|---|---|---|---|---|---|
| Small & 1x384x6 | | 0.183 | 100% | 84.3 | 76.0 | 51.0 |
| Base & 1x768x12 | $8192 \times 32$ | 0.164 | 100% | 84.7 | 78.5 | 51.8 |
| Base & 3x768x12 | | 0.145 | 95% | 84.7 | 77.9 | 51.9 |
| Base & 6x768x12 | | 0.136 | 77% | 84.6 | 63.0 | 50.1 |
| Base & 3x768x12 | $8192 \times 16$ | 0.145 | 100% | 84.7 | 76.7 | 51.7 |
| | $8192 \times 64$ | 0.148 | 67% | 84.7 | 77.6 | 51.6 |

patch aggregation. Otherwise, we consider the [CLS] token as the global representation. Table 2 presents the top-1 accuracy for linear probing and compares BEIT V2 with recent methods including BEIT, CAE, MAE, MVP and MoCo v3. All the compared methods are based on ViT-B/16 and pretrained for 300 epochs except MAE for 1600 epochs. BEIT V2 respectively outperforms BEIT, CAE and MVP by 23.4%, 16.0% and 4.7%. BEIT V2 also outperforms MoCo v3, which learns a global representation through a contrastive learning fashion. The comparisons indicate that the representation models learned by BEIT V2 enjoy higher adaptation capability.

**Robustness evaluation.** We evaluate the robustness of BEIT V2 on various ImageNet validation sets, *i.e.*, ImageNet-Adversarial (Hendrycks et al., 2021b), ImageNet-Rendition (Hendrycks et al., 2021a) and ImageNet-Sketch (Wang et al., 2019). As shown in Table 3, compared with MAE (He et al., 2022), BEIT V2 achieves dramatic gains across datasets, demonstrating the superiority of the proposed method in terms of model generalization.

### 3.3 SEMANTIC SEGMENTATION

Semantic segmentation is a dense prediction task, which generates class label for each pixel of the input image. Following the setting proposed in BEIT (Bao et al., 2022), we conduct experiments on ADE20K benchmark (Zhou et al., 2019), which includes 25K mages and 150 semantic categories. We use UperNet (Xiao et al., 2018) task layer and fine-tune the model for 160K iterations with the input resolution $512 \times 512$. Refer to Appendix G for details. Table 1 shows that BEIT V2 significantly outperforms previous self-supervised methods. Moreover, using the ViT-L/16 model, the performance can reach 56.7, which builds a new state-of-the-art for masked image modeling on ADE20k.

### 3.4 ANALYSIS

**Visual tokenizer training.** We investigate the impact of VQ-KD on BEIT V2 in terms of the model architecture and codebook size and report the results in Table 4. ViT-B/16 without the patch aggregation strategy is used as the baseline model, which is pretrained for 300 epochs. As shown in Table 4, we find that a deeper decoder of VQ-KD obtains better reconstruction, but lower codebook usage and downstream task performance. Reducing dimension for codebook lookup improves codebook utilization (Yu et al., 2021).

**Patch aggregation strategy.** Table 5 presents the ablation studies of the patch aggregation strategy. The shallower head (i.e., 1/2-layer) performs better than the deeper head (i.e., 3-layer), suggesting the shallower head pays more attention to the input [CLS] token than the deeper head. Moreover, the proposed method outperforms the baseline variant without patch aggregation strategy. The improvement of linear probe indicates better image-level representations. In addition, the results indicate that sharing the MIM head improves downstream performance.

Table 5: Ablation studies for patch aggregation strategy. $l$-**th Layer** denotes patch tokens from the $l$-th layer of the backbone. **Head Depth** means the patch aggregation head depth. **Shared MIM Head** means whether we share the MIM head parameters or not. Default settings are in gray .

| $l$-th Layer | Head Depth | Shared MIM Head | ImageNet Fine-tuning | ImageNet Linear Probe | ADE20k |
|---|---|---|---|---|---|
| *Without patch aggregation* | | | | | |
| - | - | - | 84.7 | 77.9 | 51.9 |
| *With patch aggregation* | | | | | |
| 9 | 2 | ✓ | **85.0** | **80.1** | 52.7 |
| 9 | 2 | ✗ | 84.8 | 79.5 | 51.9 |
| 9 | 1 | ✓ | 84.8 | 78.9 | 51.7 |
| 9 | 3 | ✓ | 84.7 | 78.1 | 52.0 |
| 6 | 2 | ✓ | 84.9 | 77.5 | **53.1** |
| 11 | 2 | ✓ | 84.5 | 69.4 | 51.8 |

Table 6: Comparisons between different VQ-KD targets. We also report the fine-tuning results of VQ-KD target models.

| VQ-KD Targets | ImageNet | ADE20k |
|---|---|---|
| *Pretrain 300 epochs* | | |
| DINO | 84.4 | 49.2 |
| CLIP | 85.0 | 52.7 |
| *Pretrain 1600 epochs* | | |
| CLIP | **85.5** | **53.1** |
| *Performance of VQ-KD target models* | | |
| DINO | 83.6 | 46.8 |
| CLIP | 84.9 | - |
| *Performance of VQ-KD encoder model* | | |
| VQ-KD encoder (CLIP as target) | 83.6 | - |

**VQ-KD targets.**    In Table 6, we report the results about VQ-KDs are trained under the supervision of DINO (Caron et al., 2021) and CLIP (Radford et al., 2021). DINO is pretrained solely on ImageNet-1k while CLIP is pretrained on 400M image-text pairs datasets in house. We also directly fine-tune the official base-size checkpoints and report the results in Table 6. One can see that when using DINO as the teacher model, BEiT v2 respectively reaches 84.4% and 49.2% on ImageNet and ADE20k, outperforming DINO itself by a large margin. When using CLIP as the teacher model, BEiT v2 can get consistent improvements, demonstrating the scalability of the proposed VQ-KD. In addition, we directly fine-tune the VQ-KD encoder on ImageNet. The results show that transfer performance of the VQ-KD encoder is lower than the teacher model. After performing masked image modeling, the pretrained model outperforms both the teacher model and the visual tokenizer encoder. It demonstrates the superiority of the proposed method for self-supervised learning.

**Visualization of codebook.**    We utilize the proposed VQ-KD to calculate discrete codes about the ImageNet-1k validation set. Image patches are grouped according to their corresponding codes. Figure 4 shows that the grouped image patches represent explicit semantics. For instance, the image patches corresponding to code 7856 are about "eyes" of human, cat, dog, fish and snake. Refer to Appendix A) for more examples. The introduction of codebook and feature quantization reduces the sensitiveness to the change of image details while facilitates exploitation of high-level semantics for representation models. VQ-KD compresses and quantizes the continuous feature values to a codebook, which constructs a discrete semantic space. The dimensionality of such a semantic space is significantly lower than that of the original continuous feature space. This reduces difficulty of masked patch reconstruction and alleviates the curse of dimensionality in the pretraining phase.

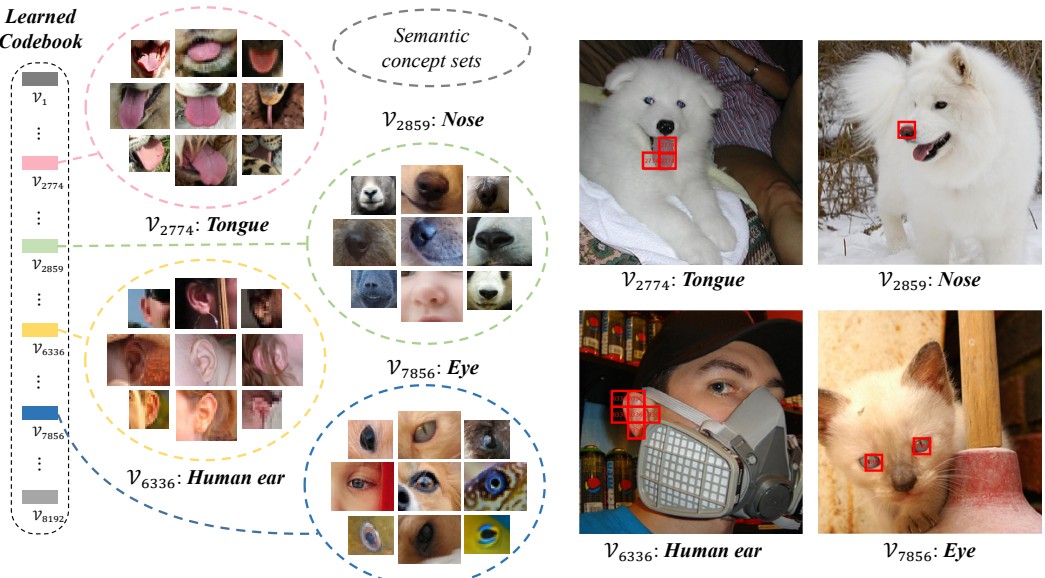

Figure 4: Visualization of semantic concepts corresponding to the learned codebook.

# 4 RELATED WORK

**Visual tokenizer.** VQ-VAE (van den Oord et al., 2017) converts an image into a sequence of discrete codes and then reconstructs the input image based on discrete codes. DALL-E (Ramesh et al., 2021) uses the Gumbel-softmax relaxation for quantization instead of the nearest neighbor lookup in VQ-VAE. VQGAN (Esser et al., 2021) and ViT-VQGAN (Yu et al., 2021) introduce Transformer block to train a better autoencoder to maintain fine details with adversarial and perceptual loss. Moreover, ViT-VQGAN proposes factorized and $\ell_2$-normalized code for codebook learning. In comparison, the proposed VQ-KD aims at reconstructing semantic knowledge from the teacher rather than original pixels. So we can construct a highly compact semantic codebook for MIM.

**Masked image modeling.** The MIM method has achieved great success in language task (Devlin et al., 2019). Motivated by it, BEIT (Bao et al., 2022) mitigated the MIM method to computer vision tasks by recovering discrete visual tokens (Ramesh et al., 2021). The prediction targets for MIM habe been explored by many recent works. MAE (He et al., 2022) treated MIM as a denoising pixel-level reconstruction task. Knowledge distillation (Wei et al., 2021; 2022) and self-distillation (Zhou et al., 2022; Baevski et al., 2022) proposed to mimic the features provided by the teacher at the masked positions. PeCo (Dong et al., 2021) regarded MoCo v3 (Chen et al., 2021) as the perceptual model in VQGAN training (Esser et al., 2021), to pursue a better tokenizer for BEIT pretraining. Despite of the progress, most existing studies remain operating on low-level image pixels, this work explores how to promote masked image modeling from pixel-level to semantic-level.

# 5 CONCLUSION

We proposed vector-quantized knowledge distillation (VQ-KD) to train a visual tokenizer for vision Transformer pretraining. VQ-KD discretized a continuous semantic space that provides supervision for masked image modeling rather than relying on image pixels. The semantic visual tokenizer greatly improved the BEIT pretraining and significantly boosted the transfer performance upon downstream tasks, such as image classification, and semantic segmentation. Moreover, a patch aggregation mechanism was introduced to explicitly encourage the model to produce global image representations, narrowing the gap between the patch-level pretraining and image-level representation aggregation. In the future, we would like to learn a universal tokenizer that projects words and images into the same vocabulary, so that we can conduct masked prediction for vision-language pretraining.

## REPRODUCIBILITY

Details of VQ-KD training, BEiT v2 pretraining, fine-tuning recipes are given in Appendix D, E, F and G. The models used for VQ-KD training are from the official repositories `https://github.com/facebookresearch/dino` and `https://github.com/openai/CLIP`. The datasets (e.g., ImageNet, and ADE20k) are derived from publicly available data buckets. The code can be found in the supplementary materials. We will also provide pretrained checkpoints to reproduce the numbers.

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

## A    VISUALIZATION OF CODEBOOK

It is observed that a discrete code tends to represent explicit semantics (Section 3.4). In Figure 5(upper), we show image examples corresponding to a given discrete code. One can see that discrete codes ignore image details, such as color, illumination, rotation and scale.

In the lower part of Figure 5, we also show some patches that mismatch the semantic concepts. Taking the fish (the first image at the last row) as instance, VQ-KD misclassifies the spot on the fish body as the eye concept due to the local structure similarity.

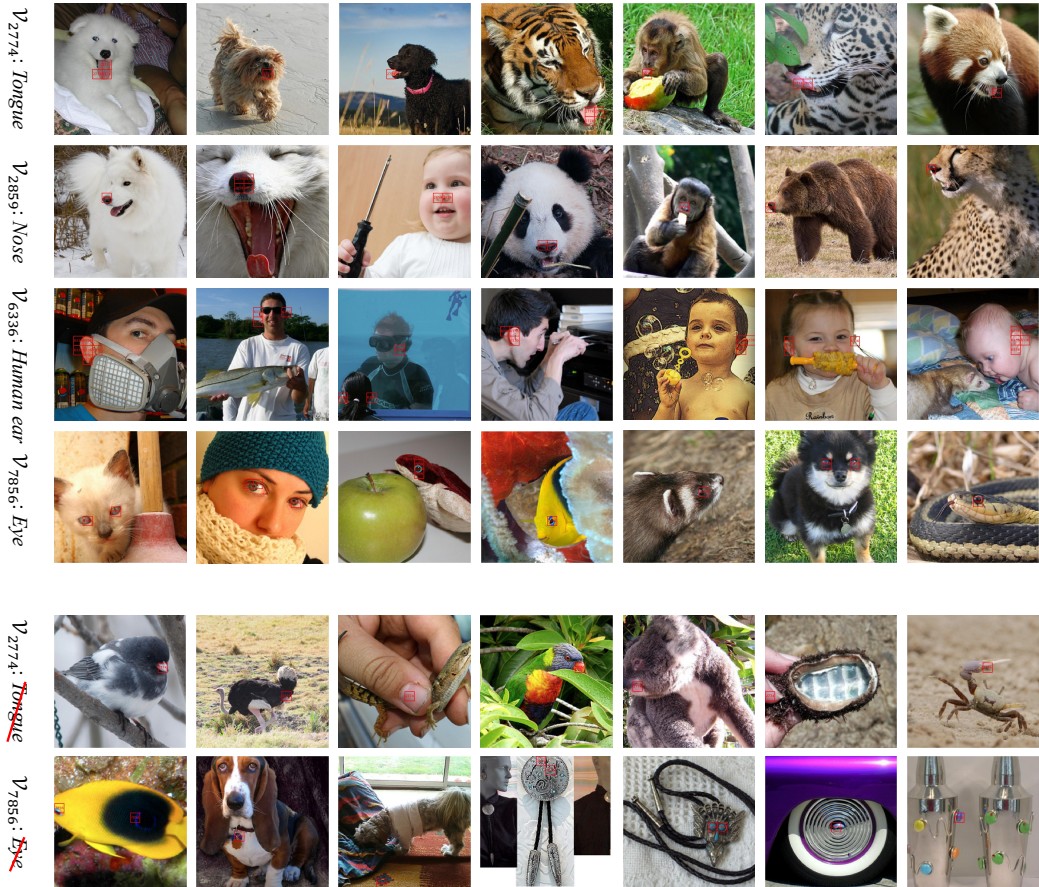

Figure 5: Visualization of image patches corresponding to discrete codes. **Upper**: examples matching the learned semantic concepts; **Lower**: patches mis-matching the semantic concepts. Corresponding patches are marked in red rectangle.

## B    COMPARISON WITH LARGE-SCALE SUPERVISED PRETRAINING

We report the performance by using the ImageNet-1k for pretraining in Table 1. To show the data scalability of BEIT v2, we conduct intermediate fine-tuning experiments on ImagNet-21k and final fine-tuning on ImageNet-1k, by using the 1600 epoch pretraining models in Table 1. From Table 7, BEIT v2 using ViT-L/16 with $384 \times 384$ input resolution, achieves 89.0% top-1 accuracy, which even outperforms ViT-H/14 using Google JFT-3B labeled dataset by 0.5%. This significant performance gain indicates the data efficiency and superiority of the proposed BEIT v2.

Table 7: Top-1 accuracy on ImageNet-1K fine-tuning. $224^2$ and $384^2$ denote model resolutions.

| Models | Model Size | Labeled Data Size | ImageNet-1k $224^2$ | $384^2$ |
|---|---|---|---|---|
| *Supervised Pretraining on ImageNet-21K* | | | | |
| ViT-B/16 (Dosovitskiy et al., 2020) | 86M | 14M | - | 84.0 |
| ViT-L/16 (Dosovitskiy et al., 2020) | 307M | 14M | - | 85.2 |
| ViT-H/14 (Dosovitskiy et al., 2020) | 632M | 14M | - | 85.1 |
| *Supervised Pretraining on Google JFT-300M (using labeled data)* | | | | |
| ViT-B/16 (Dosovitskiy et al., 2020) | 86M | 300M | - | 84.2 |
| ViT-L/16 (Dosovitskiy et al., 2020) | 307M | 300M | - | 87.1 |
| ViT-H/14 (Dosovitskiy et al., 2020) | 632M | 300M | - | 88.0 |
| *Supervised Pretraining on Google JFT-3B* | | | | |
| ViT-B/16 (Zhai et al., 2021) | 86M | 3000M | - | 86.6 |
| ViT-L/16 (Zhai et al., 2021) | 307M | 3000M | - | 88.5 |
| BEIT *Pretraining on ImageNet-21K, and Intermediate Fine-Tuning on ImageNet-21K* | | | | |
| BEIT ViT-B/16 (Bao et al., 2022) | 86M | 14M | 85.2 | 86.8 |
| BEIT ViT-L/16 (Bao et al., 2022) | 307M | 14M | 87.4 | 88.4 |
| BEIT v2 *Pretraining on ImageNet-1K, and Intermediate Fine-Tuning on ImageNet-21K* | | | | |
| BEIT v2 ViT-B/16 (ours) | 86M | 14M | 86.5 | 87.5 |
| BEIT v2 ViT-L/16 (ours) | 307M | 14M | **88.4** | **89.0** |

## C  OVERALL FRAMEWORK FOR BEIT V2

We show the tokenizer training part and BEIT v2 pretraining part in Figure 2 and Figure 3, respectively. In addition, we present the whole pretraining process in Figure 6.

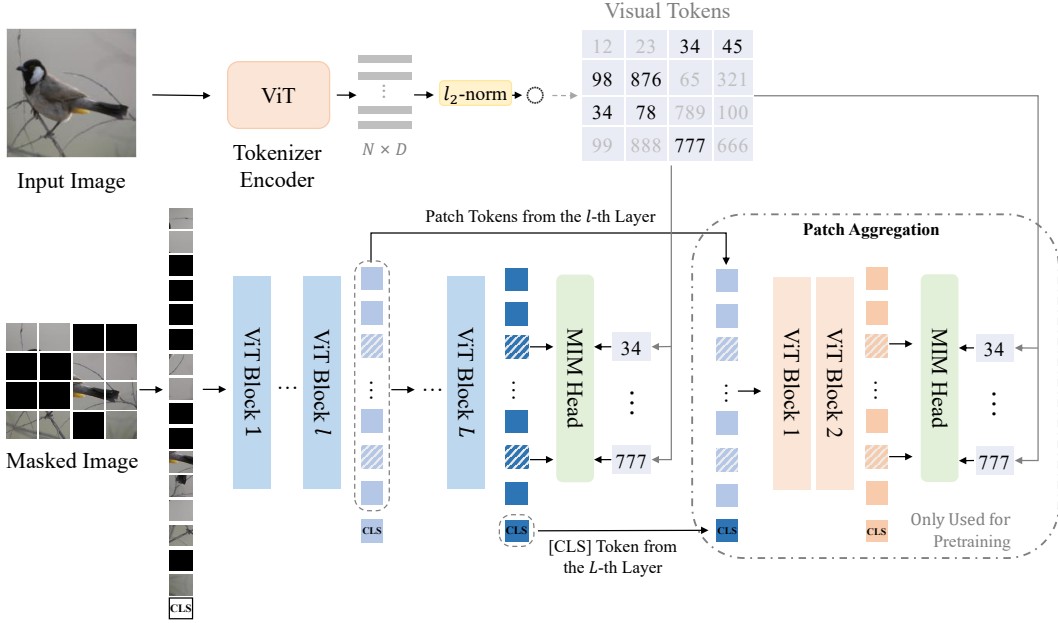

Figure 6: Overall framework for BEIT v2 pretraining.

## D  HYPERPARAMETERS FOR VQ-KD TRAINING

Table 8: Hyperparameters for training VQ-KD on ImageNet-1K.

| Hyperparameters | Values |
|---|---|
| Encoder layers | 12 |
| Decoder layers | {1, 3} |
| Hidden size | 768 |
| FFN inner hidden size | 3072 |
| Attention heads | 12 |
| Attention head size | 64 |
| Patch size | $16 \times 16$ |
| Codebook size | $8192 \times 32$ |
| Training epochs | 100 |
| Batch size | 512 |
| Adam $\beta$ | (0.9, 0.99) |
| Peak learning rate | 2e-4 |
| Minimal learning rate | 1e-5 |
| Learning rate schedule | Cosine |
| Warmup epochs | 5 |
| Gradient clipping | ✗ |
| Dropout | ✗ |
| Stoch. depth | ✗ |
| Weight decay | 1e-4 |
| Data Augment | RandomResizeAndCrop |
| Input resolution | $224 \times 224$ |

## E  HYPERPARAMETERS FOR BEIT V2 PRETRAINING

Table 9: Hyperparameters for BEIT V2 pretraining on ImageNet-1K. * denotes that the hyperparameters are adopted when the pretraining schedule is 300 epochs.

| Hyperparameters | Base Size | Large Size |
|---|---|---|
| Layers | 12 | 24 |
| Hidden size | 768 | 1024 |
| FFN inner hidden size | 3072 | 4096 |
| Attention heads | 12 | 16 |
| Layer scale | 0.1 | 1e-5 |
| Patch size | $16 \times 16$ | |
| Relative positional embeddings | ✓ | |
| Shared relative positional embeddings | ✓ | |
| Training epochs | 300*/1600 | |
| Batch size | 2048 | |
| Adam $\beta$ | (0.9, 0.98*/0.999) | |
| Peak learning rate | 1.5e-3 | |
| Minimal learning rate | 1e-5 | |
| Learning rate schedule | Cosine | |
| Warmup epochs | 10 | |
| Gradient clipping | 3.0 | |
| Dropout | ✗ | |
| Drop path | 0*/0.1 | |
| Weight decay | 0.05 | |
| Data Augment | RandomResizeAndCrop | |
| Input resolution | $224 \times 224$ | |
| Color jitter | 0.4 | |

# F HYPERPARAMETERS FOR IMAGE CLASSIFICATION FINE-TUNING

Table 10: Hyperparameters for fine-tuning BEIT V2 on ImageNet-1K.

| Hyperparameters | ViT-B/16 | ViT-L/16 |
|---|---|---|
| Peak learning rate | 5e-4 | 5e-4 |
| Fine-tuning epochs | 100 | 50 |
| Warmup epochs | 20 | 5 |
| Layer-wise learning rate decay | 0.65 | 0.8 |
| Batch size | 1024 | |
| Adam $\epsilon$ | 1e-8 | |
| Adam $\beta$ | (0.9, 0.999) | |
| Minimal learning rate | 1e-6 | |
| Learning rate schedule | Cosine | |
| Repeated Aug | ✗ | |
| Weight decay | 0.05 | |
| Label smoothing $\varepsilon$ | 0.1 | |
| Stoch. depth | 0.1 | 0.2 |
| Dropout | ✗ | |
| Gradient clipping | ✗ | |
| Erasing prob. | 0.25 | |
| Input resolution | $224 \times 224$ | |
| Rand Augment | 9/0.5 | |
| Mixup prob. | 0.8 | |
| Cutmix prob. | 1.0 | |
| Relative positional embeddings | ✓ | |
| Shared relative positional embeddings | ✗ | |

# G HYPERPARAMETERS FOR ADE20K SEMANTIC SEGMENTATION FINE-TUNING

Table 11: Hyperparameters for fine-tuning BEIT V2 on ADE20K.

| Hyperparameters | ViT-B/16 | ViT-L/16 |
|---|---|---|
| Input resolution | $512 \times 512$ | |
| Peak learning rate | {0.5, 0.8, 1.0}e-4 | |
| Fine-tuning steps | 160K | |
| Batch size | 16 | |
| Adam $\epsilon$ | 1e-8 | |
| Adam $\beta$ | (0.9, 0.999) | |
| Layer-wise learning rate decay | {0.75, 0.8, 0.85} | |
| Minimal learning rate | 0 | |
| Learning rate schedule | Linear | |
| Warmup steps | 1500 | |
| Dropout | ✗ | |
| Stoch. depth | 0.1 | 0.2 |
| Weight decay | 0.05 | |
| Relative positional embeddings | ✓ | |
| Shared relative positional embeddings | ✗ | |

