# OpenReview forum: "BEiT v2: Masked Image Modeling with Vector-Quantized Visual Tokenizers"
_ICLR.cc/2023/Conference — Submitted to ICLR 2023_

### Official Review · Reviewer_3zyP · 2022-10-21

**Confidence:** 5
**Correctness:** 4
**Technical Novelty And Significance:** 3
**Empirical Novelty And Significance:** 2
**Recommendation:** 6

**Clarity, Quality, Novelty And Reproducibility:**

Quality: good.
Clarity: good.
Originality: fair.

**Strength And Weaknesses:**

# Strength
## 1. VQ-KD
The VQ-KD follows the [1] using the l2 normalization and VITs to get codebook embeddings. To explore the high-level semantics, the VQ-KD reconstructs the feature of DINO or CLIP. The high-level semantics help the BEIT v2 have better performance.
[1] 1.  Yu, Jiahui, et al. "Vector-quantized image modeling with improved vqgan." arXiv preprint arXiv:2110.04627 (2021).
## 2. Global semantic representation
The BEIT v2 tries to use the [CLS] token to learn the global semantic representation. Combining the global representation, the BEIT v2 may have the ability to understand the global information of the image, which is essential for image classification or other tasks.
## 3. Performance
The performances in image classification and semantic segmentation are good.

# Weakness

## 1. VQ-KD
###  First question
How to explain the BEIT v2 beyond the teacher models as shown in Table [6]. I think the upper limit of BEIT v2 is teacher models, since the reconstruction loss in VQ-KD can not be zero, there has a gap between the feature provided by the VQ-KD and the teacher.

###  Second question
The authors may try RQ-VAE [2] as your basic model in VQ-KD. The paper extends the capability of codebook in channel dim, which may help the BEITv2 performance better.

[2]. Lee, Doyup, et al. "Autoregressive Image Generation using Residual Quantization." Proceedings of the IEEE/CVF Conference on Computer Vision and Pattern Recognition. 2022.

## 2. Global semantic representation
It is hard to say the [CLS] token to learn the global semantic representation in your experiments. As shown in Table [5], selecting the suitable $l$-th layer is essential to the performance. I think this is because the model not only learns the [CLS] token but also refines the other tokens in the patch aggregation strategy. In this situation, the patch aggregation strategy will let the layers($0$-th~$l$-th) predict better tokens (not the [CLS] token), and the [CLS] token is not important. The patch aggregation strategy seems like a kind of multi-scale loss and add the MIM loss in the selected layers.

So if the authors want to let the [CLS] token learn the global information better, I think you can stop the gradient computation of other tokens and just refine the [CLS] token.



**Summary Of The Paper:**

This paper proposes a new masked image modeling method(BEIT v2). A new Vector-Quantized Knowledge Distillation helps the BEIT v2 explore the high-level semantics. Meanwhile, this paper introduces a patch aggregation strategy to enhance global semantic representation. Experiments on image classification and semantic segmentation show the good performances of BEIT v2.


**Summary Of The Review:**

The BEIT v2 has good performance in image classification and segmentation. But I still have some questions as mentioned above, and I am glad to see the responses of the authors. If the authors can resolve my questions, I will increase my rating.

---

> ### Author Response · Authors · 2022-11-16
> **Response to Reviewer 3zyP (W1)**
>
> We thank the reviewer for carefully reviewing our manuscript and positive comments on the novelty and effectiveness of method.
>
> 1. *How to explain the BEIT v2 beyond the teacher models as shown in Table [6]. I think the upper limit of BEIT v2 is teacher models, since the reconstruction loss in VQ-KD can not be zero, there has a gap between the feature provided by the VQ-KD and the teacher.*
>
> **RE**: It is an interesting question why BEiT v2 can outperform the corresponding teacher.
> The reason is that BEiT v2 taps the potential of masked image modeling.
>
> As you say, the reconstruction loss in VQ-KD can not be zero, so CLIP ViT-B/16 is the upper bound of  VQ-KD tokenizer (ViT-B/16). In fact, the performance of VQ-KD tokenizer rapidly drops to 83.6\% (Table 6 in the paper). However, in this process, we can obtain a semantic-aware codebook, which is the goal of VQ-KD.
> In this codebook, most codes tend to represent one specific semantic concept, such as tongue, nose and eye, as illustrated in Figure 4 of the paper.
> That is, VQ-KD tokenizer can map the various image patches but contains the same semantic to a specific code, just like clustering.
> Remarkably, it bridges the gap between vision and language in terms of information density, where languages are human-generated signals that are highly semantic and information-dense and images are natural signals with heavy spatial redundancy (a missing patch can be recovered from neighboring patches based on lines, countours, colors and so on).
> That is, VQ-KD can reduce the spatial redundancy as much as possible, e.g., summarizing the whole ImageNet dataset in a low-dimensional yet semantic-aware codebook (8192*32).
>
> It is well known that masked language modeling appears to induce sophisticated language understanding and has achieved unprecedented success in the era of NLP.
> When the architecture gap (Transformer vs. ViT) and target gap (text tokenizer vs. VQ-KD visual tokenizer) are bridged, ViT can enjoy the gain of masked image modeling.
> This mask-then-predict strategy encourages the ViT model to have comprehensive image understanding and favorable context modeling capability, which provide a preferred initialization for various downstream tasks.
>
> In this framework, CLIP ViT-B/16 only acts as teacher to distill VQ-KD tokenizer, not acts as the teacher to pretrain BEiT v2 models.
> Using the mask-then-predict strategy, so that BEiT v2 ViT-Large model can achieve 87.3\% (Table 1 in the paper), which significantly outperforms VQ-KD tokenizer (83.6\%, Table 6 in the paper) and VQ-KD tokenizer's teacher (84.9\%, Table 6 in the paper).
>
> 2. *The authors may try RQ-VAE [2] as your basic model in VQ-KD. The paper extends the capability of codebook in channel dim, which may help the BEITv2 performance better.*
>
> **RE**: We are very grateful for this suggestion, and will add RQ-VAE [2] to related work.
> We are willing to leave it for the future work.
> As mentioned above, the visual tokenizer plays an important role in self-supervised learning (masked image modeling) task, so it's necessary to explore how to construct a better one.
>
> [1] Vector-quantized image modeling with improved vqgan.
>
> [2] Autoregressive Image Generation using Residual Quantization.

---

> ### Author Response · Authors · 2022-11-16
> **Response to Reviewer 3zyP (W2)**
>
> We thank the reviewer for carefully reviewing our manuscript and positive comments on the novelty and effectiveness of method.
>
> 3. *It is hard to say the [CLS] token to learn the global semantic representation in your experiments. As shown in Table [5], selecting the suitable $l$-th layer is essential to the performance. I think this is because the model not only learns the [CLS] token but also refines the other tokens in the patch aggregation strategy. In this situation, the patch aggregation strategy will let the layers($0$-th~$l$-th) predict better tokens (not the [CLS] token), and the [CLS] token is not important. The patch aggregation strategy seems like a kind of multi-scale loss and add the MIM loss in the selected layers.
> So if the authors want to let the [CLS] token learn the global information better, I think you can stop the gradient computation of other tokens and just refine the [CLS] token.*
>
> **RE**: According to your suggestion, we conduct the ablation experiment that detaching the gradients of patch tokens and report the results in the below Table. The pretraining epoch is 300 here.
>
> When detaching the gradients of patch tokens and only refining the [CLS] token, the model achieves 84.6\% fine-tuning accuracy and 80.0\% linear probing accuracy. Remarkably, the linear probing accuracy is 80.0\%, which is comparable to the BEiT v2, indicating the [CLS] token has learned the global semantic information as except. But for fine-tuning accuracy, it drops to 84.6\%, which is only comparable to the baseline. It also verifies your guess that a kind of multi-scale loss improves the fine-tuning accuracy.
>
> | Method                                   | Optimizing patch tokens  | Optimizing [CLS] token | Fine-tuning  acc (\%)  | Linear probing  acc (\%)  |
> |:----------------------------------------:|:------------:|:-----------:|:-----------:|:--------------:|
> | Baseline (BEiT v2 w/o patch aggregation) | $\times$     | $\times$    | 84.7        | 77.9           |
> | Baseline + optimizing [CLS] token        | $\times$     | $\surd$     | 84.6        | 80.0           |
> | BEiT v2                                  | $\surd$      | $\surd$     | 85.0        | 80.1           |

---

> ### Author Response · Authors · 2022-11-19
> **Does our response answer your questions?**
>
> We thank the reviewer for the great effort in reviewing our manuscript. Does our response answer your questions? If not, we are willing to discuss further.

---

> ### Author Response · Authors · 2022-12-09
> **Looking forward to post-rebuttal discussions**
>
> Dear Reviewer 3zyP,
>
> We would like to express our sincere gratitude for your time and efforts in reviewing our paper. Your comments and feedback will be invaluable in helping us improve the final version of our paper.
>
> As the deadline for discussion is approaching, we are willing to provide any additional clarifications that you may require. In our previous response, we have carefully considered your comments and made detailed responses, which are summarized below:
> + We explained why BEIT v2 outperforms the teacher models.
> + We are willing to leave your suggestions for the future work.
> + We conducted an ablation experiment that detaching the gradients of patch tokens as you suggested.
>
> We hope the above experiments and analyses could clarify your concerns. Thanks for your time and efforts!
>
> Best,
>
> Authors.

---

### Official Review · Reviewer_d8pL · 2022-10-22

**Confidence:** 4
**Correctness:** 4
**Technical Novelty And Significance:** 2
**Empirical Novelty And Significance:** 2
**Recommendation:** 5

**Clarity, Quality, Novelty And Reproducibility:**

- This paper is  well written with sufficient details, so it may not be diffcult to reproduce.

**Strength And Weaknesses:**

Strengths
- The idea of introducing a semantic-rich visual tokenizer into the BEIT framework is reasonable and interesting.
- This paper is clearly written and easy to follow.

Weaknesses
- Using a teacher model for supervising the visual tokenizer training makes the whole pretraining procedure more complex and time consuming. Two additional training stages (teacher model training and tokenizer training) are required before the final masked image modeling pretraining when compared to other one-stage pretraining methods, which weakens the significance of this work. A detailed comparision of the GPU hours (number of used GPUs and training Days) of different training stages of BEITv2 and other competitors may help clarify this issue.
- The final pretraining performance (evaluated with imagenet finetuning) strongly relies on the teacher model. As shown in Table 6, better teacher model (CLIP vs DINO) can consitently bring better performance of BEITv2. While, BEITv2 can improve DINO from 83.6 to 84.4, but can only improve CLIP from 84.9 to 85.0, which indicates that with stronger teacher, the performance gain of BEITv2 will rapidly decrease. When Patch Aggregation strategy is not used, the performance of BEITv2 even drops to 84.7 (Table 4), lower than the CLIP teacher, which further weakens the significance of this work.
- The novelty of of introducing Patch Aggregation strategy is somewhat limited as it shows little difference from Condenser Head in Gao & Callan 's work (Condenser: a Pre-training Architecture for Dense Retrieval, 2021). The architecture and the usage (used for pretraining but dropped during fine-tuning) are all similar.

**Summary Of The Paper:**

This paper improves BEIT by introducing the Vector-Quantized Knowledge Distillation (VQ-KD) algorithm for better visual tokenizer training. A patch aggregation strategy is also introduced into the Masked-Image-Modeling (MIM) pretraining framework. Experimental results show that BEITv2 (this work) significantly outperforms other MIM pretraining methods.

**Summary Of The Review:**

I mainly worry about the significance of the propsed method, considering the strong teacher required and more complex training stages. The novelty of the Patch Aggregation strategy is also limited. The authors are encouraged to address the above issues.

---

> ### Author Response · Authors · 2022-11-16
> **Response to Reviewer d8pL (W1)**
>
> We thank the reviewer for carefully reviewing our manuscript.
>
> 1. *Using a teacher model for supervising the visual tokenizer training makes the whole pretraining procedure more complex and time consuming. Two additional training stages (teacher model training and tokenizer training) are required before the final masked image modeling pretraining when compared to other one-stage pretraining methods, which weakens the significance of this work. A detailed comparision of the GPU hours (number of used GPUs and training Days) of different training stages of BEITv2 and other competitors may help clarify this issue.*
>
> **RE**: We'd like to report the GPU hours of each stage of BEiT v2 for a detailed comparison in the below Table: 1) For teacher model training, we just use the official public pretrained model checkpoints, so it's hard to present the detailed training time; 2) For tokenizer training, it spends about 373 GPU hours (23 hours on 16 V100s) in our settings (using ViT-Base model and training for 100 epochs); 3) As for MIM pretraining, it spends about 3908 GPU hours (244 hours on 16 V100s) when pretraining ViT-Base for 1600 epochs and about 18773 GPU hours (293 hours on 64 v100s) when training ViT-Large for 1600 epochs.
> Therefore, compared to the MIM pretraining stage, the time consuming on tokenizer training can be ignore (9.5\% for ViT-Base, 1.9\% for ViT-Large).
> Remarkably, the most important point is that once the tokenizer is determined, it can be employed in all the MIM pretraining experiments, just like the tokenizer in NLP.
>
> | Stage                       | GPU Hours (on one V100) | Ratio of tokenizer training to MIM pretraining |
> |:---------------------------:|:-----------------------:|:----------------------------------------------:|
> | Tokenizer training          | 373                     | -                                              |
> | MIM pretraining (ViT-Base)  | 3908                    | 9.5\%                                          |
> | MIM pretraining (ViT-Large) | 18773                   | 1.9\%                                          |
>
> To present a more comprehensive comparison, we compare BEiT v2 with two popular methods, MAE [1] and data2vec [2], in terms of pretraining time. MAE proposes to reconstruct removed pixels at masked positions. data2vec proposes to reconstruct features from the EMA teacher at masked positions.
> We test all models on a 4 V100 machine with the same settings. The results can be found in the below Table,where average step time is calculated from (total training time) / (total steps). MAE and data2vec are evaluated by using their official codebase.
>
> Benefiting greatly from the decoupling encoder-decoder architecture, MAE shows  advantages in terms of training efficiency but suffers from low performance.
> That is, MAE requires more pretraining time for the similar performance with BEiT v2.
> Compared to data2vec, BEiT v2 shows the superiority on both training efficiency and performance.
> One reason is that EMA teachers are almost the same size as their students, while VQ-KD can relax this constraint.
>
> | Architecture | Method   | Supervision       | Average step time (s) | MIM epochs | Accuracy (\%) |
> |:------------:|:--------:|:-----------------:|:---------------------:|:----------:|:--------:|
> | ViT-Base     | MAE      | Normalized pixels | 0.272                 | 1600       | 83.6     |
> | ViT-Base     | data2vec | EMA features      | 0.636                 | 800        | 84.2     |
> | ViT-Base     | BEiT v2  | VQ-KD tokens      | 0.605                 | 1600       | 85.5     |
> | ViT-Large    | MAE      | Normalized pixels | 0.290                 | 1600       | 85.9     |
> | ViT-Large    | data2vec | EMA features      | 1.01                  | 1600       | 86.6     |
> | ViT-Large    | BEiT v2  | VQ-KD tokens      | 0.705                 | 1600       | 87.3     |
>
> Although constructing a visual tokenizer for MIM is more complex than using pixels as target, VQ-KD enjoys scalability and high performance.
> Moreover, VQ-KD shows the superiority in vision-language pretraining.
> In BEiT-3 [3], visual codebook and language codebook are combined as the supervision for masked data modeling.
> The unity greatly balances the gradients and stables the pretraining procedure in a large-scale dataset.
>
> [1] Masked Autoencoders Are Scalable Vision Learners
>
> [2] data2vec: A General Framework for Self-supervised Learning in Speech, Vision and Language
>
> [3] Image as a Foreign Language: BEIT Pretraining for All Vision and Vision-Language Tasks

---

> ### Author Response · Authors · 2022-11-16
> **Response to Reviewer d8pL (W2 & W3)**
>
> We thank the reviewer for carefully reviewing our manuscript.
>
> 2. *The final pretraining performance (evaluated with imagenet finetuning) strongly relies on the teacher model. As shown in Table 6, better teacher model (CLIP vs DINO) can consitently bring better performance of BEITv2. While, BEITv2 can improve DINO from 83.6 to 84.4, but can only improve CLIP from 84.9 to 85.0, which indicates that with stronger teacher, the performance gain of BEITv2 will rapidly decrease. When Patch Aggregation strategy is not used, the performance of BEITv2 even drops to 84.7 (Table 4), lower than the CLIP teacher, which further weakens the significance of this work.*
>
> **RE**: Tables 4 and 6 in the paper are our ablation experiments with the 300 epochs pretraining schedule.
> We summarize the results in the below Table.
> BEiT v2 can consistently outperform VQ-KD's teacher (83.6\%).
> When pretraining ViT-Base for 300 epochs, BEiT v2 can achieve 85.0\%, which is slightly higher than CLIP B/16 (85.0\% vs. 84.9\%).
> However, when extending the schedule to 1600 epochs, BEiT v2 can boost to 85.5\% accuracy.
> When replacing ViT-Base with ViT-Large, BEiT v2 can reach 87.3\% accuracy, which is significantly better than CLIP B/16 (87.3\% vs. 84.9\%).
> Those results show the generalization and scalability of VQ-KD.
>
> | Teacher    | Student    | Pretraining epochs | Accuracy |
> |:----------:|:----------:|:------------------:|:--------:|
> | -          | CLIP-B/16  | N/A                | 84.9\%   |
> | CLIP-B/16  | VQ-KD Base | 100                | 83.6\%   |
> | VQ-KD Base | ViT-Base   | 300                | 85.0\%   |
> | VQ-KD Base | ViT-Base   | 1600               | 85.5\%   |
> | VQ-KD Base | ViT-Large  | 1600               | 87.3\%   |
>
> 3. *The novelty of of introducing Patch Aggregation strategy is somewhat limited as it shows little difference from Condenser Head in Gao & Callan 's work (Condenser: a Pre-training Architecture for Dense Retrieval, 2021). The architecture and the usage (used for pretraining but dropped during fine-tuning) are all similar.*
>
> **RE**: We introduce the patch aggregation strategy to form the global image representation, inspired by Condenser.
> Although the similar method is under-explored in masked language modeling, we provide extensive experimental results and show the potential for masked image modeling, which is our unique contribution.
> We still believe it is open and interesting to explore how to generate a global image representation in the patch-level proxy task (i.e., masked image modeling).
> Contrastive learning could be an important approach to explicitly generate the global representation and has been extensively studied in the past years, but it suffers from heavy data augmentation and large batch size.
> Therefore, it is nontrivial to provide an alternative strategy.
> In this work, we construct a information bottleneck to implicitly encourage CLS token to collect patch features as much as possible.
> CLS token then can be treated as the global image representation for image-level task, like linear probing.
> Moreover, the enhanced representations also facilitate other downstream tasks, like semantic segmentation.

---

> ### Author Response · Authors · 2022-11-19
> **Does our response answer your questions?**
>
> We thank the reviewer for the great effort in reviewing our manuscript. Does our response answer your questions? If not, we are willing to discuss further.

---

> ### Author Response · Authors · 2022-12-09
> **Looking forward to post-rebuttal discussions**
>
> Dear Reviewer d8pL,
>
> We would like to express our sincere gratitude for your time and efforts in reviewing our paper. Your comments and feedback will be invaluable in helping us improve the final version of our paper.
>
> As the deadline for discussion is approaching, we are willing to provide any additional clarifications that you may require. In our previous response, we have carefully considered your comments and made detailed responses, which are summarized below:
> + We provided the detailed comparison of GPU hours of different stages and other competitors.
> + We highlighted the generalization and scalability of VQ-KD when extending the pretraining schedule.
> + We demonstrated our contribution to forming a global image representation.
>
> We hope the above experiments and analyses could clarify your concerns.  Thanks for your time and efforts!
>
> Best,
>
> Authors.

---

### Official Review · Reviewer_jP3q · 2022-10-24

**Confidence:** 3
**Correctness:** 3
**Technical Novelty And Significance:** 3
**Empirical Novelty And Significance:** 3
**Recommendation:** 6

**Clarity, Quality, Novelty And Reproducibility:**

The paper is clear, high quality, and has significant novelty. The authors included the code as supplementary material for reproducibility.

**Strength And Weaknesses:**

**Strengths**
- The two contributions are well justified and the extensive
experimental results strongly validate the design decisions.

- The method outperforms compared pre-training methods by
  significant margins across diverse tasks.

- The obtained representations are significantly more robust compared
  to the state-of-the-art pretraining method MAE.

**Weaknesses/Questions**
- Unless I missed it, it would be nice to explain how the projection
  for reducing the code dimension for lookup is obtained.
- In (2), there is no $\beta$ multiplying the commitment term as in the
  standard VQ-VAE, but isn't $\beta=1$ equivalent to
  directly optimizing the distance without stop-gradient? If
  $\beta$ was used, please specify its value.
- I'm curious about how the clusters in Figure 4 where found. Was it
  by visual inspection of occurrences of all (or many) of the codebook
  elements? Have you compared this type of semantic association with the
  baseline tokenizer?


**Summary Of The Paper:**

This paper proposes an improved version of the BEiT method for
self-supervised representation learning by predicting masked image
patches. The improvements rest mainly in two incorporations:
1) Encoding image patches using a ViT-based tokenizer that distillis the
semantic information from a CLIP model into the encoded image tokens.
2) Adding a patch aggregation stage that makes class token aggregate
   global information from the patch encodings.

Extensive experiments show the effectiveness of the improvements in
standard ImageNet pre-training benchmarks for classification and
ADE-20K for semantic segmentation.


**Summary Of The Review:**

The paper proposes methodologic contributions that significantly
advance the state-of-the-art  in large-scale pretraining for computer
vision tasks.  The contributions are well justified and the paper present
sufficient empirical evidence through extensive experiments.

---

> ### Author Response · Authors · 2022-11-16
> **Response to Reviewer jP3q**
>
> We thank the reviewer for carefully reviewing our manuscript and positive comments on the novelty and effectiveness of method.
>
> 1. *Unless I missed it, it would be nice to explain how the projection for reducing the code dimension for lookup is obtained.*
>
> **RE**: This trick is proposed in ViT-VQGAN [1] to achieve high codebook usage for reconstructing images.
> When VQ-KD encoder is ViT-B/16 and the codebook size is 8192\*32, we use a fully-connection layer to map features (768-dim) to a lower dimension (32-dim).
>
> 2. *In (2), there is no $\beta$ multiplying the commitment term as in the standard VQ-VAE, but isn't  $\beta=1$ equivalent to directly optimizing the distance without stop-gradient? If $\beta$ was used, please specify its value.*
>
> **RE**: In experiments, we disable the commitment loss by updating the codebook with the EMA approach, which is proposed in vanilla VQ-VAE [2].
> That is, Equation 2 can be rewritten as:
> $$\mathrm{max} \sum_{x \in D} \sum_{i=1}^N\cos{(o_i , t_i )}- ||\mathrm{sg}[\ell_2(h_i )] - \ell_2(v_{z_i})||_2^2.$$
>
> 3. *I'm curious about how the clusters in Figure 4 where found. Was it by visual inspection of occurrences of all (or many) of the codebook elements? Have you compared this type of semantic association with the baseline tokenizer?*
>
> **RE**: You are right.
> As discussed in the Section 3.4, we utilize VQ-KD to calculate discrete codes on the whole ImageNet-1k validation set (not used during the VQ-KD training), and then group all patches based on their corresponding code index.
> After that, most codes tend to represent one specific semantic concept by visual inspection.
> Unfortunately, it lacks a quantitative protocol to measure this phenomenon.
> As you see, the cluster in VQ-KD is obvious and interesting.
> Some concepts can be described in text, such as the ear, nose, and eye, while some not, like conjunctions in language.
> According to your suggestion, we conduct the same visualization strategy on dVAE tokenizer [3] (used in BEiT), but unfortunately do not find such a phenomenon.
>
> [1] Vector-quantized image modeling with improved vqgan.
>
> [2] Neural Discrete Representation Learning.
>
> [3] Zero-Shot Text-to-Image Generation.

---

> ### Author Response · Authors · 2022-11-19
> **Does our response answer your questions?**
>
> We thank the reviewer for the great effort in reviewing our manuscript. Does our response answer your questions? If not, we are willing to discuss further.

---

### Official Review · Reviewer_Y8Gd · 2022-10-25

**Confidence:** 4
**Correctness:** 2
**Technical Novelty And Significance:** 3
**Empirical Novelty And Significance:** 3
**Recommendation:** 5

**Clarity, Quality, Novelty And Reproducibility:**

 - Major clarity and presentation is clear, where as many typos exist: e.g. in Table 5: l-th Layer denotes **path** tokens.

 - While some interesting techniques are adopted in this work, it's unclear where the actual improvement comes from. Therefore, the novelty is still questionable.

**Strength And Weaknesses:**

#### **Strength**

 - Masked Image modeling and pre-training has been an important topic in computer vision. Different from predicting pixels directly (MAE-style), this work promotes the development of token prediction (BEiT/BERT-style) pre-training.

 - The authors conduct a variety of experimental studies. With the introduced changes, promising results on image classification and semantic segmentation are shown.

 - Linear probing has been a relatively weak point of MIM-based pre-training, compared to contrastive and self-distillation methods. However, this work upgrades the model's linear performance.

#### **Weakness**

 - Albeit with promising performance, the major issue in this work is about the use of CLIP features. With VQ-KD training, the encoder outputs could be viewed as distillations from CLIP, which is further leveraged to guide the MIM training. This paradigm results in unfair comparison with previous works. (See summary)

 - The vision encoder of CLIP already shows a 76.2 zero-shot performance and 84.9(reported in Table 6) fine-tuned accuracy. Considering the significance of its raw features, it remains questionable of the value about such modified pre-training targets. With additional efforts and computation spent on VQ-KD and MIM pre-training, the final accuracy is only improved marginally. Given the context that proper knowledge distillation could further improve fine-tuned accuracy [1], the benefits of MIM pre-training are hindered.

 - Several tricks in ViT-VQGAN are adopted in this work. While it's non-trivial to exploit advances in generative modeling, there lacks discussion over these modifications such as L2 norm, and factorize codes (e.g. from 256-dim to 32-dim). Since ViT-VQGAN have already conducted studies on representation learning and shows improvement in linear probing. The technical contribution seems to be limited here.

 - As expressed in the paper, the patch aggregation strategy is designed to obtain enhanced global representations.  However, from the comparison in Table 1 and Table 5, the design choice seems to be adhoc and most performance elevation comes from the CLIP-distilled codebook. The role of patch aggregation lacks further investigation.

[1] Contrastive Learning Rivals Masked Image Modeling in Fine-tuning via Feature Distillation.

**Summary Of The Paper:**

 - This work extends the study of BEiT, i.e. BERT-style pre-training of image transformers. Specifically, this work proposes to adopt a semantic-rich visual tokenizer distilled from the semantically-rich CLIP model, promoting the MIM process to focus more on semantic-level.

 - In order to learn better and compact codebook space, this work also exploits other practices adopted in previous vector-quantized modeling paper, such as factorized code, and L2 norm. Meanwhile, a patch aggregation strategy is introduced in MIM to enhance global semantic space.

 - Extensive experiments on image classification and semantic segmentation are conducted, which demonstrates the effectiveness of this method.

**Summary Of The Review:**

 - As promising results are provided in this work, it's also valuable to design better vector quantization modules for representation learning. However, the unfair comparisons and marginal improvement (comparing to directly using CLIP ) suspect the contribution of this work.

---

> ### Author Response · Authors · 2022-11-16
> **Response to Reviewer Y8Gd (W1 & W2)**
>
> We thank the reviewer for carefully reviewing our manuscript.
>
> 1.*Albeit with promising performance, the major issue in this work is about the use of CLIP features. With VQ-KD training, the encoder outputs could be viewed as distillations from CLIP, which is further leveraged to guide the MIM training. This paradigm results in unfair comparison with previous works.*
>
> **RE**: One of the most important topics of pretraining is how to seek and utilize useful information for downstream tasks. In our work, we tap the potential of masked image modeling by using the public resource (such as datasets and model weights). From a broader view of self-supervised learning, one of our contributions is show that the usage of CLIP features can bring significant gains than without using them. In a high-level sense, the comparisons are still fair, especially our goal is to get good-quality pretrained vision models.
>
> Moreover, in this work, VQ-KD is encouraged to reconstruct the features from a pretrained teacher model.
> The teacher model is flexible here.
> In order to compare with previous works, we use DINO and CLIP as VQ-KD's teachers in our experiments.
> When using DINO as VQ-KD's supervision, BEiT v2 can outperform MaskFeat [3] by 0.4\% accuracy (84.4\% vs. 84.0\%) on ImageNet-1k with the 300 epochs pretraining schedule. MaskFeat [3] directly predicts the DINO's features, so it's a fair comparison in terms of supervision.
> When using CLIP as VQ-KD's supervision, we compare BEiT v2 with MVP [4] in Table 1 of the paper, where MVP [4] directly predicts CLIP features.
> Specifically, BEiT v2 surpasses MVP by 0.6\% accuracy and 0.3 mIoU when pretraining the ViT-Base model, and exceeds MVP by 0.3\% accuracy and 0.7 mIoU when pretraining the ViT-Large model.
>
> As a conclusion, under the comparable setting, BEiT v2 achieves the superiority to previous works across various datasets and model sizes.
>
> 2. *The vision encoder of CLIP already shows a 76.2 zero-shot performance and 84.9(reported in Table 6) fine-tuned accuracy. Considering the significance of its raw features, it remains questionable of the value about such modified pre-training targets. With additional efforts and computation spent on VQ-KD and MIM pre-training, the final accuracy is only improved marginally. Given the context that proper knowledge distillation could further improve fine-tuned accuracy [1], the benefits of MIM pre-training are hindered.*
>
> **RE**: Compared to feature distillation methods, BEiT v2 obtains better performance and scalability.
> Our point is that masked image modeling can go beyond feature distillation.
> For example, the performance of student in FD-CLIP[1] after distilling is 84.9\%, which is comparable to the teacher we fine-tuned (84.9\%; notably, we use the official model checkpoint without any modification). In comparison, our method achieves 85.5\% (1600 epochs), which is higher than distillation.
>
> Furthermore, we conduct the comparison experiment on that using CLIP ViT-B/16 to distill ViT-L/16 on each patch according to FD-CLIP[1], and then the student achieves 85.4\% accuracy (300 epochs), which is a marginal gain.
> One possible reason may be that the large student is easy to fully reconstruct the small teacher's latent space, causing the marginal gain compared to CLIP ViT-B/16.
> However, under the comparable recipe, BEiT v2 using ViT-L/16 can reach 86.6\% accuracy (Table 1 in the paper), indicating that a base-size VQ-KD tokenizer can drive a large-size model for MIM pretraining.
>
> [1] Contrastive Learning Rivals Masked Image Modeling in Fine-tuning via Feature Distillation.
>
> [2] Vector-quantized image modeling with improved vqgan.
>
> [3] Masked Feature Prediction for Self-Supervised Visual Pre-Training
>
> [4] Multimodality-guided Visual Pre-training

---

> ### Author Response · Authors · 2022-11-16
> **Response to Reviewer Y8Gd (W3 & W4)**
>
> We thank the reviewer for carefully reviewing our manuscript.
>
> 3. *Several tricks in ViT-VQGAN are adopted in this work. While it's non-trivial to exploit advances in generative modeling, there lacks discussion over these modifications such as L2 norm, and factorize codes (e.g. from 256-dim to 32-dim). Since ViT-VQGAN have already conducted studies on representation learning and shows improvement in linear probing. The technical contribution seems to be limited here.*
>
> **RE**: In this work, VQ-KD is encouraged to reconstruct the features from a pretrained teacher model, not the pixel values as ViT-VQGAN [2] do.
> We focus on building a semantic-aware visual tokenizer for MIM, not for generative modeling.
> In fact, we have tried to remove the L2 norm during the code index lookup but have encountered the codebook collapse problem (i.e., only few codes are activated), which makes it non-meaningful to perform the subsequent MIM pretraining.
> In Table 4 of the paper, we have discussed the impact of factorized codes. When decreasing the codebook size from 8192\*64 to 8192\*32, the codebook usage is boosted, and the performance achieves consistent improvements.
> Despite using prior techniques to stabilize codebook training, we believe that the contribution of VQ-KD is to provide a more straightforward method to construct a semantic-aware visual tokenizer for MIM task.
>
> 4. *As expressed in the paper, the patch aggregation strategy is designed to obtain enhanced global representations. However, from the comparison in Table 1 and Table 5, the design choice seems to be adhoc and most performance elevation comes from the CLIP-distilled codebook. The role of patch aggregation lacks further investigation.*
>
> **RE**: As demonstrated in Table 5 of the paper, when the information bottleneck is suitably constructed, BEiT v2 enjoys significant improvement in terms of the linear probing metric, as well as the semantic segmentation task.
> The goal of the patch aggregation strategy is to mitigate the discrepancy between patch-level pretraining and image-level representation aggregation.
> Moreover, it provides an alternative method to form competitive global representation, compared to contrastive learning.
>
> [1] Contrastive Learning Rivals Masked Image Modeling in Fine-tuning via Feature Distillation.
>
> [2] Vector-quantized image modeling with improved vqgan.

---

> ### Author Response · Authors · 2022-11-18
> **Response to Reviewer Y8Gd (Clarity)**
>
> *Major clarity and presentation is clear, where as many typos exist: e.g. in Table 5: l-th Layer denotes path tokens.*
>
> **RE**: Thank you for your meticulous review. We have fixed it in the revised manuscript.

---

> ### Author Response · Authors · 2022-11-19
> **Does our response answer your questions?**
>
> We thank the reviewer for the great effort in reviewing our manuscript. Does our response answer your questions? If not, we are willing to discuss further.

---

> ### Author Response · Authors · 2022-12-09
> **Looking forward to post-rebuttal discussions**
>
> Dear Reviewer Y8Gd,
>
> We would like to express our sincere gratitude for your time and efforts in reviewing our paper. Your comments and feedback will be invaluable in helping us improve the final version of our paper.
>
> As the deadline for discussion is approaching, we are willing to provide any additional clarifications that you may require. In our previous response, we have carefully considered your comments and made detailed responses, which are summarized below:
> + We addressed the fairness concern that existed in comparing BEiT v2 with previous works.
> + We added the experiment to compare BEiT v2 with the paper you mentioned. Moreover, when intermediate fine-tuning is performed on ImageNet-21k, BEiT v2 reaches 89.0% accuracy, which surpasses that of ViT-Large trained on the JFT-3B dataset. This indicates that using CLIP features in this work does not hinder the benefits of MIM pretraining but rather enhances them.
> + We highlighted the differences between ViT-VQGAN and VQ-KD.
> + We highlighted that BEiT v2 enjoys significant improvement when the information bottleneck is suitably constructed.
> + We corrected the typo that you pointed out.
>
> We hope the above experiments and analyses could clarify your concerns.  Thanks for your time and efforts!
>
> Best,
>
> Authors.

---

### Decision · Program_Chairs · 2023-01-20

**Decision:**

Reject

**Justification For Why Not Higher Score:**

The paper primarily compares to prior SSL methods that learn from unlabeled images alone; however the method distills from a pretrained CLIP model, so implicitly uses much more supervision than SSL baselines.

**Justification For Why Not Lower Score:**

N/A

**Metareview: Summary, Strengths And Weaknesses:**

The paper proposes an extension to BEiT, which uses masked image modeling as a pretext task for pre-training Vision Transformers (ViTs). The paper proposes to train an image tokenizer to reconstruct the features of a teacher model (CLIP or DINO), which is then used during masked image modeling (MIM) pretraining of the ViT. The paper also proposes a new mechanism for patch aggregation in ViTs.

Reviewer responses to this paper were mixed, with Reviewers Y8Gd and d8pL giving slightly negative ratings, Reviewer 3zyP giving a slightly positive rating, and only Reviewer jP3q strongly in favor of the paper. Reviewers were concerned that the proposed method did not outperform the CLIP teacher by a wide margin, wanted more details on the training cost of the method, and had questions about the patch aggregation strategy. The authors provided extensive responses to all reviewer questions, but this was not able to fully satisfy the reviewers; in the post-rebuttal discussion Reviewer Y8Gd remained unconvinced by the author’s responses, and Reviewer 3zyP felt that the paper was borderline.

The AC met with Reviewers jP3q, Y8Gd, and 3zyP to discuss this paper. The major issue we discussed was the positioning of the paper, and the validity of using a pretrained CLIP teacher as part the pretraining procedure. Reviewers were not satisfied with the relatively small improvement over the CLIP teacher model. In addition, we discussed the issue of supervision and baselines: the paper compares primarily to prior work that trains with only *unlabeled* images; however since the proposed method uses a pretrained CLIP model, it implicitly has access to large quantities of image/text supervision. As such, for fair comparison the method should be compared to baselines that have access to the same level of supervision. All reviewers agreed that these were serious issues, and following this discussion all reviewers in attendance agreed that the paper is not ready for publication.

**Summary Of Ac-Reviewer Meeting:**

The AC met with Reviewers jP3q, Y8Gd, and 3zyP to discuss this paper. We primarily discussed the use of a pretrained CLIP model as discussed above.